# Application of Zebrafish Model in the Suppression of Drug-Induced Cardiac Hypertrophy by Traditional Indian Medicine Yogendra Ras

**DOI:** 10.3390/biom10040600

**Published:** 2020-04-13

**Authors:** Acharya Balkrishna, Yashika Rustagi, Kunal Bhattacharya, Anurag Varshney

**Affiliations:** 1Drug Discovery and Development Division, Patanjali Research Institute, Haridwar 249 401, India; pyp@divyayoga.com (A.B.); yashika.rustagi@prft.co.in (Y.R.); 2Department of Allied and Applied Sciences, University of Patanjali, Patanjali Yog Peeth, Haridwar 249 401, India

**Keywords:** Ayurveda, cardiac hypertrophy, zebrafish model, H9C2, erythromycin, isoproterenol, oxidative stress, Yogendra Ras, biomarkers

## Abstract

Zebrafish is an elegant vertebrate employed to model the pathological etiologies of human maladies such as cardiac diseases. Persistent physiological stresses can induce abnormalities in heart functions such as cardiac hypertrophy (CH), which can lead to morbidity and mortality. In the present study, using zebrafish as a study model, efficacy of the traditional Indian Ayurveda medicine “Yogendra Ras” (YDR) was validated in ameliorating drug-induced cardiac hypertrophy. YDR was prepared using traditionally described methods and composed of nano- and micron-sized metal particles. Elemental composition analysis of YDR showed the presence of mainly Au, Sn, and Hg. Cardiac hypertrophy was induced in the zebrafish following a pretreatment with erythromycin (ERY), and the onset and reconciliation of disease by YDR were determined using a treadmill electrocardiogram, heart anatomy analysis, C-reactive protein release, and platelet aggregation time-analysis. YDR treatment of CH-induced zebrafish showed comparable results with the Standard-of-care drug, verapamil, tested in parallel. Under in-vitro conditions, treatment of isoproterenol (ISP)-stimulated murine cardiomyocytes (H9C2) with YDR resulted in the suppression of drug-stimulated biomarkers of oxidative stress: COX-2, NOX-2, NOX-4, ANF, troponin-I, -T, and cardiolipin. Taken together, zebrafish showed a strong disposition as a model for studying the efficacy of Ayurvedic medicines towards drug-induced cardiopathies. YDR provided strong evidence for its capability in modulating drug-induced CH through the restoration of redox homeostasis and exhibited potential as a viable complementary therapy.

## 1. Introduction

The zebrafish (*Danio rerio*) has emerged as a useful tool in the study of human diseases, recapitulating some or all of the observed pathologies. Through whole-genome mapping, *D. rerio* has been replacing small animal models in understanding disease modalities and their conserved molecular pathways. *D. rerio* has also been used as a model for studying cardiac stress and diseases using electrocardiograms (ECGs) and biochemical parameters, since they closely resemble those of humans [1,2,3]. The heart is a major organ involved in the perfusion of blood and oxygen demands to the distal organs [4]. During cardiac failures, it is unable to adequately pump blood to the organs in response to systemic demands and can lead to mortality [5]. Cardiac hypertrophy (CH) is induced through physical and pathological stress, which produces stimuli for the cardiomyocytes to grow in length and width. Physiologically, this leads to an increase in the cardiac pump function while decreasing ventricular wall tension and in turn inducing compensated CH. It is also accompanied by an increase in the left ventricular wall thickness, as a response to the reduction of systolic and diastolic stress on the left cardiac wall. Long-term persistence of CH could lead to heart failure, arrhythmia, and sudden death [6].

Erythromycin (ERY) is a macrolide antibiotic generally associated with gastrointestinal distress. ERY is also a motilin receptor agonist that reduces the G-coupled receptor-stimulated contractions in the smooth muscle. Studies have shown that oral treatment of patients with ERY followed by its metabolism by cytochrome P450 3A (CYP3A) leads to an increase of the drug concentration in the blood plasma, consequently leading to the development of cardiac arrhythmia and torsade de pointes (TdP) by blocking human-ether-a-go-go gene (hERG) and prolonging QTc intervals [7]. In *D. rerio*, ERY has been successfully shown to be an inducing agent of CH and TdP [8,9]. Isoproterenol (ISP) is a synthetic catecholamine and stimulator (agonist) for β-adrenergic receptors (β-AR) that exerts its effect on the cardiac tissues. The sustained activation of the β-AR and an increase in the cytosolic Ca^2+^ levels can lead to the induction of CH, producing the hypertrophic phenotype [10,11]. Cardiomyocytes are mainly composed of mitochondria due to high energy requirements and generate reactive oxygen species (ROS) as a by-product of oxidative phosphorylation [12,13]. Therefore, the progression of CH has also been associated with the induction of oxidative stress, mitochondrial membrane depolarization, and apoptosis [12].

Recently, there has been a growing interest in the treatment of diseases with traditional Ayurvedic medicines [14,15,16,17]. Yogendra Ras (YDR) is an ancient, traditional Indian metal-based formulation that has been prepared using Ras Sindoor (herbally processed sulfur and mercury ash), 28%; Loha bhasma (herbally processed iron ash); Swarna bhasma (herbally processed gold ash); Abhrak bhasma (herbally processed mica ash); Shuddha Mukta (a powdered form of pearl ash); Vang bhasma (herbally processed tin ash), 14% each; along with *Aloe vera* leaves juice as the binding agent. This Ayurvedic formulation is prepared as per the procedures described in the several century-old ancient Indian medicinal texts of *Bhaiṣajya Ratnāvalī* and *Vatavyadhi Chikitsa*, for healing neuropathological, cardiovascular and diabetes diseases [18,19]. In *Bhaiṣajya Ratnāvalī* (Vātavyādhyādhikāra; 506-512), the sloka (in sanskrit) mentions “Viśuddham rasasindūram taddwardham śuddhahāṭakam. Tatsamam kāntalauhñca tatsamañcābhrameva ca. Viśuddham mauktikañcaiva vaṅgañca tatsamam matam. Kumarikārasairbhāvayam dhānyarāśau dinatrayam” (Translation: Herbally processed mercury (Rasasindūra), gold (Hāṭaka) ash (Bhasma), iron (Kāntalauha) ash, mica (Abhra) ash, pearl (Mauktika) ash, tin (Vaṅga) ash are to be mixed in *Aloe vera* juice (Kumārī rasa) and triturated (bhaāvanā) in a vessel (kharala). It is then mixed under pressure (Mardāna) in Kumārī rasa; made into pellets and dried. The pellets are to be wrapped with *Ricinus communis* (Eraṇḍa) leaves and kept in a heap of dry *Oryza sativa* (Dhānya) seeds for 3 days, before use). The prescribed human dosage of the YDR according to *Bhaishajya Ratnavali* is 125 mg twice a day [18]. Though there are some concerns regarding the safety effects of pure Hg, in the ancient Indian and Chinese medicinal systems, Hg containing herbally processed medicines have been prescribed for healing skin, heart, and neurological diseases [20,21].

In the current study, we set up a *D. rerio* CH model, measured cardiac electrical activities, and tested the efficacy of YDR in protecting against drug (ERY)-induced CH. Parameters measured were modulation of electrocardiogram PQRST waves, change in heart size, production of C-reactive protein (CRP), and platelet aggregation. The mode of action for the YDR was studied under in-vitro conditions in the ISP-stimulated murine cardiomyocytes (H9C2) by measuring parameters such as the intonation of oxidative stress, mitochondrial dysfunction, and clinical and non-clinical cell signaling biomarkers for cardiac function.

## 2. Materials and Methods

### 2.1. Source of Test Compounds and Reagents

Yogendra Ras (YDR) (batch number A-YGR011) was procured from Divya Pharmacy, Haridwar, India, sold under its classical name. Based on sensory characterizations, the YDR formulation was observed to be a dry, free-flowing powder and rust-brown in color, packed under inert environment. The YDR powder in dried condition was odorless, tasteless, and insoluble in water and cell culture media due to its metal-based origin under normal physiological pH and temperature. Reagents for cell culture studies such as Dulbecco’s modified Eagle medium (DMEM), fetal bovine serum (FBS), antibiotics, trypsin-EDTA, isoproterenol, 2′,7′–dichlorofluorescein diacetate (DCFDA), MitoTracker red dye, TaqMan primers and universal RT-PCR master mix for quantitative real-time PCR analysis were purchased from Thermo Fisher Scientific Inc., USA. Purified catalase enzyme was purchased from Merck India Pvt Ltd. Purified superoxide dismutase standards were purchased from Cayman Chemicals, USA. Trichloroacetic acid and Giemsa stain were purchased from Hi-media Laboratories, India. Verapamil was purchased from Sigma-Aldrich, India, and erythromycin was purchased from TCI Chemicals, India.

### 2.2. Physicochemical Analysis of the Yogendra Ras (YDR)

The morphological and elemental attributes of the dry YDR powder were measured using a scanning electron microscope (SEM; LEO-438 VP) with an attached electron dispersive X-ray analysis system (Carl Zeiss, Germany). Gold sputtering of the YDR sample to improve the electron micrograph quality was done at an accelerated voltage of 10 kV. Powder X-ray diffraction (XRD) analysis was performed using a Rigaku D-Max 2200 X-ray diffractometer applying Cu-Kα radiation at 40 kV/40 mA. Scanning was performed throughout the experiment at a step width of 0.02° over an angular range of 5° to 80° and a scanning rate of 0.5° min^−1^.

For particle size distribution analysis, the YDR sample was suspended individually at a concentration of 100 mg/mL in double-distilled water and in the DMEM cell culture media containing 2% FBS. Measurements were done in triplicates using the Malvern Zetasizer Nano ZS (Malvern Panalytical, United Kingdom). Inductively coupled mass spectroscopy (ICP-MS) analysis of the YDR samples was performed at the Eurofins Analytical Services India Private Limited, Bengaluru, India.

### 2.3. Experimental Animals: D. rerio Maintenance

*D. rerio* obtained from the in-house breeding facility at Pentagrit Research, Chennai, India were used in the current study. A total of 216 adult male *D. rerio* with a bodyweight of 0.5 g and length of 25-30 mm were selected for this study. The fish were divided randomly into two groups and acclimatized as 12 fish per polycarbonate tank containing 2 L of water, prior to the effective and therapeutic dose studies. Throughout the study duration, a period of light (14 h) and dark (10 h) cycle and a constant water temperature of 27 ± 1 °C were maintained. The fish were fed TetraMin^®^ flakes obtained from Tetra, VA. Fish were divided into 9 groups containing 24 fish per group. All the experiments were performed following the protocol approved by the Institutional Animal Ethics Committee (IAEC) in accordance with the Committee for the Purpose of Control and Supervision of Experiments (CPCSEA), Government of India (approval number: 222/Go102019/IAEC), and were in general compliance with the ARRIVE guidelines [22].

#### 2.3.1. Step 1: Cardiac Hypertrophy Induction in *D. rerio*

Inducer stock of erythromycin was prepared by dissolving it at the concentration of 2 mg/mL in 0.9% saline solution and was stored at −20 °C until further use. For the induction of CH, 50 µL of the stock solution (100 µg of erythromycin) was added to 4 L of the housing water of *D. rerio*. All the erythromycin-exposed fish were maintained for a period of 16 days in case of an effective dose study and for 23 days in case of a therapeutic dose study (Figure 1). In the normal control fish, an equivalent volume of 0.9% saline solution was added to the housing water. Normal control and erythromycin-exposed fish were observed and stabilized from day 5 onwards for the initiation of YDR/verapamil treatments.

#### 2.3.2. Step 2: Dosing of Test Articles in *D. rerio*

YDR doses for *D. rerio* were optimized at 1000× less than the relative human doses (125 mg/day BID) by body weights [23,24]. Hence, the doses selected for the “effective dose screening” in *D. rerio* were 0.6, 4, and 18 µg/kg (Figure 1). Standard-of-care cardiac drug, verapamil was given to the fish at the concentration of 4 µg/kg (human equivalent dose). For oral exposure, the selected amount of YDR formulation was mixed with the known mass of the fish feed (TetraMin^®^) and was extruded into uniform pellets. For feeding individual fish, a rectangular fish tank was separated into 6 independent units, and individual fish were separated from the respective study groups. Each fish was fed individually on a 24 h cycle with an estimated number of pellets, under isolated condition. Control fish were fed with unmodified fish feed under conditions similar to the exposed groups.

### 2.4. Treadmill Electrocardiography Analysis

Unlike the human heart, *D. rerio* heart is comprised of a single ventricle and a single atrium. The cardiac functions are cyclic with every systole where the ventricle pumps blood into the bulbus arteriosus; this acts as a reservoir from which blood empties into the ventral aorta [25]. In *D. rerio*, the heart is anatomically located right in the midline of the ventral side, immediately below the gills. The bulbus arteriosus is present on the dorso-cranial side of the ventricle. The ventricle apex is directed to the trunk. To measure the treadmill ECG, the study fish were transferred to experimental treadmill tanks as described by Depasquale et al. [26]. The treadmill tank is a glass tank consisting of 6 chambers of uniform size to study 6 samples at a time. Each chamber has a water inlet and outlet drain pipe. The water flow rate was maintained at 230 L/h. Fish were aerobically challenged to swim in the controlled water flow rate and water velocity. A maximum velocity of 0.5 m/sec was used in the treadmill chambers for fish. Fish were first introduced to a pre-treatment chamber, which is similar to the treadmill chamber. The water flow rate and water velocities were 23 L/h and 0.05 m/sec, respectively, in the pre-treatment chamber; this prevents sudden anxiety and bias in measurements. The fish were allowed to acclimatize for 5 min in the pre-treatment chamber and were shifted to the treadmill chamber. Fish were exposed in the treadmill chamber for 3 min each, after which the ECG was assessed.

Three-channel ECGs were used to measure the electrocardiograph cardiac signatures [27]. The channels were distributed as two on the head and one to the caudal region. In the head, channels were placed on either side of the gills closer to the heart. The caudal channel was placed on the region where the trunk connects to the caudal region. The ECG was recorded at 20 mV and at a speed of 5 mm/sec. Observations were recorded, and ECG graphs were analyzed statistically. Induction of CH was calculated on the basis of Cornell product, Cornell voltage, Sokolow–Lyon, and Romhilt–Estes point criteria [28].

### 2.5. Measurement of Platelet Aggregation and C-Reactive Protein (CRP)

Fish were sedated in cold water set at 14 °C. A small slit was made injuring the major blood vessel in the tip caudal region of the fish using a sharp scalpel. Care was taken to keep the fish alive until the clotting time was recorded. The cut was approximately 0.25 mm deep passing through the scales and the skeletal muscle. Blood was collected and immediately spotted on a glass slide. Whole blood was observed under a light microscope at 4×, and the platelet aggregation was measured as a parameter for the blood clotting time.

After measurement of the platelet aggregation, all the study fish were euthanized immediately using ice-cold water set at 4 °C. Blood was collected in a glass tube containing 5 µL of 0.5% EDTA by making an incision in the region of the dorsal aorta and inferior vena cava, just posterior to the dorsal fin near the caudal region. EDTA-treated whole blood was then centrifuged at 10,000 RPM for 10 min to separate the plasma. Collected plasma was then stored at −80 °C until further use, and CRP levels were measured using the Randox CRP analysis kit and auto-analyzer system (Randox Laboratories Ltd., United Kingdom).

### 2.6. Post-Mortem Heart Anatomy

Heart was dissected out from the euthanized fish. The isolated heart was immersed and stored in 30% formalin solution for 72 h. Post-fixative treatment of the heart was dissected in the cross-sectional direction into 3 parts. The middle section was used to measure the septal wall thickness. Microscopic images of the heart were taken at 4× magnification and the thickness was measured using Image J software.

### 2.7. In-Vitro Cell Culture

The H9C2 (rat embryonic cardiac) cells were purchased from the ATCC licensed cell repository National Centre for Cell Science (NCCS), Pune, India. The H9C2 cells were cultured in DMEM media supplemented with 10% fetal bovine serum (FBS) and 1% antibiotics. Cells were cultured at the density of 2 × 10^6^ cells/cm^2^ and grown in a humidified incubator at 37 °C with 5% CO_2_. For the experiments, cells were plated in 96-well plates at the density of 10,000 cells/well. The cells were pre-incubated before exposure to the stimulant and the drugs.

### 2.8. Cell Viability Assay

H9C2 cells were treated with ISP or YDR mixed in fresh cell culture media in semi-log doses between 0 and 1000 µg/mL for 24 h. At the end of the exposure time, the old medium was removed, and cells were washed with 100 μL PBS. A total of 100 μL of 0.5 mg/mL 3-(4,5-dimethylthiazol-2-yl)-2,5-diphenyltetrazolium bromide (MTT) was added to each well, and the plates were incubated for 3 h at 37 °C. At the end of the exposure period, 75 μL of MTT dye was removed. Fifty microliters of DMSO was added, and the plates were placed on a shaker for 10 min. The absorbance of each well was read using the PerkinElmer Envision microplate reader at 595 nm wavelength, and cell viability percentage was calculated along with an inhibitory concentration of 20% (IC_20_) and 50% (IC_50_).

### 2.9. Determination of Intracellular Oxidative Stress

H9C2 cells were exposed to ISP (50 µM) and YDR (30 µg/mL) individually or in combination for 24 h. At the end of the exposure period, cells were washed with lukewarm PBS, and fresh media containing 10 µg/mL of 2′,7′-dichlorofluorescein diacetate (DCFH-DA) dye was added to all the wells. The plates were incubated in the dark for 45 min at 37 °C. Fluorescence intensity was measured using the PerkinElmer Envision microplate reader at the excitation and emission wavelengths of 490 and 520 nm, respectively.

### 2.10. Mitochondrial Membrane Potential Assay

H9C2 cells were exposed to ISP (50 µM) and YDR (30 µg/mL) individually or in combination for 24 h. At the end of the exposure period, the cells were washed with lukewarm PBS and stained with 1 mM of MitoTracker Red probes for 45 min. Cells were washed with 1× HBSS buffer after removing staining media, and fluorescence intensity was measured on the PerkinElmer Envision microplate reader with excitation and emission at 584 and 606 nm, respectively.

### 2.11. Nitroblue Tetrazolium Assay for Superoxide Generation

H9C2 cardiomyocytes were seeded in the 6-well culture plate, and after 60–70% confluence, cells were treated with ISP and YDR each for 24 h. After the completion of the exposure, 0.3% NBT was added to each well and incubated for 1 h in an incubator at 37 °C with 5% CO_2_. Cells were counterstained with 2% safranin for 10 min. Stained cells were observed under an inverted microscope at 10× for the formation of blue-purple formazan crystals. The formazan crystals were solubilized with 2 M KOH and 0.5 N HCl to calculate the percentage of NBT reduction [29]. Elution was taken out in a 96-well plate, and absorbance was measured at 630 nm in a plate reader. The stimulation index was calculated by the ratio of absorbance of treated and control cells.

### 2.12. Isolation and Quantification of Total Cell Proteins

H9C2 cells were exposed to ISP (50 µM) and YDR (30 µg/mL) individually or in combination for 24 h. At the end of the exposure period, the cells were washed with lukewarm PBS, trypsinized, and lysed in RIPA buffer NaCl (150 mM), NP-40 (10% *v*/*v*), Tris-HCl (50 mM; pH 7.5), EDTA (1 mM), PMSF, and protease inhibitor cocktail and incubated for 1 h on ice with intermittent tapping. The lysate was then centrifuged at 13,000 rpm for 10 min at 4 °C. Protein concentration was quantified in the collected supernatant using a BCA kit (Thermo Fisher Scientific Inc., USA) as per the manufacturer instructions. Unused supernatant was stored at −80 °C for further analysis.

### 2.13. Catalase Enzyme Assay

Catalase activity was measured in isolated proteins as per the method mentioned by Weydert et al. [30]. Initially, absorbance was adjusted between 1.150 and 1.200 by diluting 300 µL of H_2_O_2_ (30% *v*/*v*) in 100 mL of potassium phosphate buffer. Then, 125 µg/mL of extracted protein was added to 0.1M potassium phosphate buffer (pH 7.0), and the reaction was initiated by adding 30 mM H_2_O_2_. Absorbance was recorded spectrophotometrically for every 30 s at a wavelength of 240 nm for 3 min in a quartz cuvette. Purified catalase was used to plot the standard curve. The enzyme activity was expressed as millimolar H_2_O_2_ consumed per min per mg of protein.

### 2.14. Superoxide Dismutase (SOD) Enzyme Assay

SOD enzyme activity was measured based on the method of Beauchamp and Fridovich using Cayman’s superoxide dismutase assay kit as per the manufacturer instructions [31]. The reaction was processed by adding 200 µL of the diluted radical detector (1.5 mM nitroblue tetrazolium chloride) in 20 µL of isolated protein, and the reaction was initiated under direct white light at interval 0 by adding 10 µL of diluted 0.12 mM riboflavin in a 96-well plate. The plate was incubated in light on a shaker for 12 min, and absorbance was recorded at 560 nm using a plate reader. Purified SOD was used to plot the standard curve, and the amount of enzyme required to inhibit the reduction of NBT by 50% and the activity was expressed as units/mL of protein.

### 2.15. Determination of mRNA Expression Level of Clinical and Non-Clinical Cardiac Biomarkers

Gene expression studies using quantitative real-time PCR (qRT-PCR) were performed to evaluate the mRNA expression level of clinical and non-clinical cardiac biomarkers in H9C2 cells. The cells were exposed to ISP (50 µM) and YDR (30 µg/mL) individually or in combination for 24 h. At the end of the exposure period, the cells were washed with lukewarm PBS and lysed using TRIZOL reagent (Thermo Fisher Scientific Inc., USA), and total RNA was isolated. RNA was purified and quantified by taking absorbance at 260 and 280 nm. Each RNA sample was processed to cDNA using a one-step Verso cDNA synthesis kit (Thermo Fisher Scientific Inc., USA). For qRT-PCR, TaqMan chemistry-based FAM single tube primer assays of NOX-2 and -4, COX-2, ANF, CLRS-1, TNN-T and TNN-I, along with TaqMan universal master mix were used. Beta-actin (ACTB) housekeeping gene was used as an internal control to normalize the expression of other genes. Real-time PCR assay was run on a Biometra TProfessional RT-PCR machine (Analytik-Jena AG, Germany), and cycling parameters included initial denaturation at 95 °C for 10 min and primer extension at 95 °C for 15 s and 60 °C for one min with 40 cycles. Ct values were obtained, relative expression 2^−ΔΔCt^ was calculated, and data were analyzed for fold change in mRNA expression.

### 2.16. Statistical Analysis

All the experiments were performed in technical triplicates as well as in biological triplicates. Data from all the experiments are presented as the mean ± SEM. An unpaired t-test was used for comparison between two groups. For groups of three or more, the data were subjected to one-way analysis of variance (ANOVA) followed by Dunnett’s post-hoc test. Differences were considered statistically significant if the *p*-value was ≤ 0.05.

## 3. Results

### 3.1. Physicochemical Properties of Yogendra Ras (YDR) Particles

Evaluation of the scanning electron microscope (SEM) images of YDR showed them to be heterogeneous in shape and size (Figure 2A). YDR was found in the particulate form either present loosely or in large clusters with diameters of < 100 nm to ~1 µm. Surface chemical characterization of the YDR using the electron dispersive X-ray (EDX) technique identified the presence of mercury (Hg) (37.75 ± 1.92%); tin (Sn) (30.05 ± 3.35%); oxygen (O) (14.90 ± 1.08%); calcium (Ca) (12.86 ± 1.99%); iron (Fe) (5.44 ± 0.45%); silicon (Si) (2.44 ± 0.26%); and traces of (< 1%) magnesium (Mg), sodium (Na), aluminum (Al), and zinc (Zn) on the surface of the YDR particles (Figure 2B,C).

Inductively coupled plasma mass spectroscopy (ICP-MS) analysis of YDR samples showed the presence of 9.8% mercury (Hg) and 7.9% gold (Au). X-ray diffraction (XRD) crystallography analysis of the YDR sample confirmed the presence of Hg in the form of Cinnabar and Metacinnabar (HgS) (Figure 3A). Using the XRD analysis, the presence of pure gold (Au), hematite (Fe_2_O_3_), arsenic trioxide (As_4_O_6_), and AsFe were identified in the YDR samples (Figure 3A). Several other crystal peaks were also detected in low amounts in the XRD analysis. Dynamic light scattering (DLS) analysis of the YDR particles showed a size distribution transformation when the suspension medium was changed from water to complete cell culture media. In pure water suspension, YDR showed a bimodal size distribution with an average dynamic size distribution of 1116.00 ± 102.70 d.nm (Figure 3B). However, when suspended in the DMEM media supplemented with 2% FBS, YDR size distribution changed to a monomodal distribution showing a size of 486.40 ± 46.95 d.nm.

Change in the size distribution of the YDR particles in the presence of the cell culture media biomolecules was further supported by the polydispersity index (PdI) values indicating that the YDR particles formed a more stable colloidal suspension in the DMEM cell culture media supplemented with 2% FBS (PdI = 0.174 ± 0.06) as compared to the particles suspended in pure water (PdI = 0.406 ± 0.080) (Figure 3B).

Anatomical analysis of the normal control *D. rerio* heart showed structurally intact bulbus arteriosus, atrium, and ventricle (Figure 4A,Bi). Heart retained a dense red color of the ventricle in the normal fish (Figure 4Bi,Ci). Heart anatomy analysis of the disease control fish post-7-day exposure to ERY showed intact bulbus, atrium, ventricle regions, and normal coloration (Figure 4Bii). However, after 14 days of exposure to ERY, significant anatomical changes representing hypertrophy were observed in the heart of the disease control fish with the appearances of dark coloration as compared to the normal control fish (Figure 4Cii). Septum wall thickness analysis showed a significant (*p*-value < 0.001) time-dependent septum wall thickness increase in the ERY-stimulated *D. rerio* in comparison to the normal control fish (Figure 5A). Measurement of the ECG parameters in the effective dose study showed the normal control fish to have a regular PQRST deflection with standard intervals and segments. The normal fish showed a heartbeat between 140 and 150 beats per minute confirming the amplitude of 0.5 millivolts (mV) deflection for the QR wave (Figure 4Bi,Ci). Induction of *D. rerio* with erythromycin (ERY; 25 µg/Lt) for both the 7-day (effective dose screening) and 14-day (therapeutic dose testing) exposure studies showed a widening in the angle of QRS complex compared to the normal control fish (Figure 4Bi,Bii,Ci,Cii). The R wave showed an increase of 5 mm, and the S wave showed a dip by 3 mm indicating an increase in the Cornell voltage criteria and an increase in wave amplitudes. The resulting increase in the time for completion of a single beat was determined to be 0.2 decaseconds compared to the normal control fish (Figure 4Bi,Bii,Ci,Cii). Additional ECG markers determined for the inception of the CH in the disease control fish were Sokolow–Lyon criteria, as observed in Cornell voltage criteria and Cornell product criteria, and Romhilt–Estes point through an ST elevation of 0.025 mV for the isoelectric segment of the time period (Figure 4Bii,Cii).

### 3.2. YDR Ameliorates Cardiac Hypertrophy and Associated Heart Anatomical Abnormalities

For the effective dose screening, the ERY-induced *D. rerio* were treated with three doses of YDR: low (0.6 µg/kg), medium (4 µg/kg), and high (18 µg/kg) (Figure 1, Figure 4Biii–v). Anatomically, a low dose of YDR + ERY-treated fish showed mild fluid infiltration and heart size increase (Figure 4Biii). The medium YDR dose + ERY-treated fish showed a bulged heart at the ventricle and atrium side and mild fluid infiltration in the bulbus region (Figure 4Biv). The highest YDR dose + ERY-treated fish showed an intact heart with bulbus, atrium, and ventricle (Figure 4Bv). Septum wall thickness analysis indicated that the YDR treatment at the low doses sustained the ERY stimulated increase in the wall thickness, whereas the medium and high dose treatments of YDR significantly ameliorated the increased septal wall thickness (Figure 5A). All the YDR-treated fish showed partial recovery from the ERY-stimulated CH following the 7-day treatment with an R wave showing a 2.5 mm decrease and S wave dip by 1.5 mm indicating a decrease in Cornell voltage criteria, as compared to the disease control fish. The decrease in the R and S waves showed a reduction in the intensity of CH stimulated by ERY. However, the presence of amplitude of 0.12 millivolts (mV) indicated the tendency to depolarize the ventricle and induction of mild CH (Figure 4Biii–v). The presence of CH in the YDR-treated fish was also confirmed as per the Cornell product criteria represented by the presence of QRS angle widening and the additional requirement of 0.2 decaseconds to complete a single beat (Figure 4Biii–v). Additional hypertrophy markers such as the Sokolow–Lyon criteria as observed in Cornell voltage criteria and Cornell product criteria and Romhilt–Estes point also confirmed the presence of CH in the 7-day YDR + ERY-treated fish, with an ST elevation of 0.0125 mV for the isoelectric segment of the time period (Figure 4Biii–v).

Anatomical analysis of ERY-induced and YDR-treated fish hearts revealed the presence of very mild hypertrophic morphology and normal morphology in the standard-of-care drug verapamil + ERY-treated fish (Figure 4Ciii,iv). Septum wall thickness analysis revealed significant recovery in the ERY-stimulated increase in the wall thickness, by treatment with the therapeutic dose of YDR or standard drug verapamil (Figure 5D). ECG analysis of the therapeutic YDR dose and standard of care drug verapamil-treated *D. rerio* showed a complete recovery of the ERY-stimulated ECG parameters related to the onset of CH, as compared to the disease control fish (Figure 4Cii–iv). All the ECG parameters such as Cornell voltage criteria, Cornell product criteria with QRS angle, Sokolow–Lyon criteria (as per Cornell voltage criteria and Cornell product criteria), and Romhilt–Estes point confirmation with no ST elevation measured in ERY + YDR and ERY + verapamil-treated fish showed parameters equal to those observed in the normal control fish (Figure 4Ci–iv).

No mortality was detected in any of the control or treatment groups in the 7-day effective dose screening study. In the therapeutic dose screening study, a total of four mortalities were observed in the disease control fish, probably due to disease severity. No mortality was detected in the YDR therapeutic dose or standard drug treatment groups.

### 3.3. YDR Inhibits Stimulated Expression of Serum C-Reactive Protein

Determination of the C-reactive protein (CRP) levels present in the blood serum of the *D. rerio* in the effective dose screening showed a significant increase in protein expression on ERY stimulation (Figure 5B). Elevated levels of CRP have been associated with increased cardiovascular risks like left ventricle hypertrophy and hypertension in children and adults [33]. Hence, in our study, elevated levels of CRP represented the development of CH in the fish following induction with ERY. Treatment of the fish with varying concentrations of the YDR formulation led to a concentration-dependent decrease in the elevated levels of CRP (Figure 5C). This reduction represents the modulation of CH by the YDR as determined using other screening parameters. In the therapeutic study, selecting the highest effective dose (18 µg/kg) of the YDR, a complete normalization in the elevated levels of CRP was detected. Interestingly, verapamil (4 µg/kg) did not reduce ERY-stimulated CRP levels. Hence, the results indicated that while verapamil was not able to reverse the acute-phase induction of CH in the ERY-treated *D. rerio*, YDR was able to completely ameliorate the drug-induced cardiac effect and underlying mechanism of action (Figure 5E).

### 3.4. YDR Effect on Stimulated D. rerio Platelet Aggregation Time

Clinical studies have shown that patients with cardiac diseases, especially ischemia and cardiomyocyte hypertrophy, tend to show high spontaneous and induced rates of platelet aggregation [34]. In the present study, ERY stimulation of the *D. rerio* led to a decrease in the platelet aggregation time as compared to the normal control fish (Figure 5C). This ERY stimulated reduction in the platelet aggregation time as a precursor to the development of thrombosis. Treatment of the ERY-stimulated *D. rerio* with different doses of YDR (0.6–18 µg/kg) significantly (*p*-value < 0.001) recovered the platelet aggregation time back to normal as compared to the disease control fish. Therapeutic treatment of the ERY-stimulated fish with YDR (18 µg/kg) and the standard drug verapamil also showed a significant recovery of the platelet aggregation compared to the disease control (Figure 5F).

### 3.5. In-Vitro Dose Screening of Yogendra Ras (YDR) and Isoproterenol (ISP) in H9C2 Cells

Cell viability analysis showed a dose-dependent change in the H9C2 cells following exposure to ISP or YDR (Figure 6A). Based on these results, the inhibitory concentrations of 20% (IC_20_) and 50% (IC_50_) for ISP were determined to be 50 and 122.4 µM, respectively. Similarly, IC_20_ and IC_50_ doses for the YDR were determined to be at the concentrations of 30 and 205 µg/mL, respectively. On the basis of this dose screening, 50 µM of ISP and 30 μg/mL of YDR were selected for the subsequent cell biological experiments.

### 3.6. Oxidative Stress Analysis

The stimulation of H9C2 cells with ISP (50 µM) for 24 h produced a significant (p-value < 0.01) 1.3-fold increase in the intracellular ROS generation (Figure 6B). Elevated levels of oxidative stress due to excessive accumulation of reactive oxygen species (ROS) have been reported to induce CH [12]. Hence, an increase in ROS indicated the stimulated generation of oxidative stress and CH within the ISP-stimulated H9C2 cells. Concurrent treatment of the ISP (50 µM)-induced H9C2 cells with varying doses of YDR significantly (1 µg/mL *p* < 0.05; 3 µg/mL *p*< 0.05; 10 µg/mL *p* < 0.01; 30 µg/mL *p* < 0.01) reduced the intracellular generation of ROS in a dose-dependent manner (Figure 6B).

Mitochondria play an important role in the production and release of intracellular ROS. Stimulation of the H9C2 cells with ISP (50 µM) led to significant (*p*-value < 0.01) escalation in the mitochondrial membrane potential (MMP) (Figure 6C). This increase in MMP is well-correlated to the elevated generation of ROS in the H9C2 cells stimulated with ISP (Figure 6B,C). Co-treatment of the ISP (50 µM)-treated H9C2 cells with YDR significantly (*p*-value < 0.01) reduced the elevated level of MMP in a dose-dependent manner (Figure 6C).

Mitochondrial origin superoxide anions (O^−2^) were detected intracellularly using a nitroblue tetrazolium assay. We observed a significant (*p*-value < 0.001) rise in the generation of intracellular O^−2^ in the H9C2 cells when treated with ISP (50 µM) alone (Figure 6D). Concurrent treatment of the ISP-induced H9C2 cells with YDR significantly (*p*-value < 0.01) reduced the stimulated release of O^−2^ (Figure 6D). These results correlated well with the observed modulation of ISP-stimulated intracellular ROS generation by YDR in the H9C2 cells (Figure 6B,C). Hence, based on these results, we could validate that YDR efficiently reduced the oxidative stress-stimulating property of the ISP in the cardiomyocytes.

### 3.7. Anti-Oxidants Modulation by YDR in ISP-Induced H9C2 Cells

Stimulation of the H9C2 cells with ISP (50 µM) led to a significant (*p*-value < 0.01) reduction in the levels of the anti-oxidants i.e., superoxide dismutase (SOD) and catalase, indicating onset of oxidative stress (Figure 6E,F). This was significantly (*p*-value < 0.01) recovered through the co-treatment of the ISP-stimulated H9C2 cells with YDR showing a recovery in the levels of both SOD and catalase. The SOD and catalase enzymatic assay result further correlated well with the previously observed high induction of O^−2^ and ROS in the H9C2 cells when treated with ISP alone and its amelioration following a co-treatment with YDR (Figure 6B,D).

ROS generation is known to induce the expression of Cyclooxygenase (COX) and the subclass of NADPH oxidases (NOX), which further act as inflammatory mediators during the induction of cardiotoxicity. Therefore, we studied the oxidative stress-stimulated up-regulation of the mRNA expression profiles of pro-inflammatory mediators COX-2, NOX-2, and NOX-4 (Figure 7A–C). Both COX-2 and NOX-4 showed a significant (*p*-value < 0.01) fold increase following the induction of H9C2 cells with ISP (50 µM), while the elevated levels of NOX-2 were not statistically significant. YDR alone did not induce any stimulated increase in the inflammatory mediators. In the present study, concurrent treatment of ISP-stimulated H9C2 cells with YDR (30 µg/mL) led to a (*p* <0.05) reduction in the expression of COX-2 and NOX-4 (Figure 7A,C). The reduction was also observed in the case of NOX-2 but was not statistically significant when compared to the ISP-alone-stimulated H9C2 cells (Figure 7B). Thus, the results validated the observed antioxidant and anti-inflammatory properties of the YDR in ameliorating ISP-stimulated oxidative stress and inflammation in the cardiomyocytes through concurrent treatment.

### 3.8. Non-Clinical and Clinical Genetic Biomarker Modulation by YDR in ISP-Induced H9C2 Cells

The onset of CH has been associated with the activation of the fetal cardiac remodeling genes. Following stimulation of the H9C2 cells with ISP, significant (*p*-value <0.01) overexpression of fetal cardiac gene atrial natriuretic factor (ANF) was observed (Figure 7D). This overexpression of the ANF was down-regulated following a co-treatment of the ISP-treated H9C2 cells with YDR (30 µg/mL) (Figure 7D). Similarly, other essential clinical biomarker genes such as troponin I (TNN-I), troponin T (TNN-T), and cardiolipin (CRLS-1) were also found to be up-regulated in the ISP-stimulated H9C2 cells as compared to normal controls. Overexpression of these clinical biomarker genes was significantly reduced in the ISP-treated H9C2 cells following a concurrent treatment with YDR (30 µg/mL) (Figure 7E–G). Expression of the clinical and non-clinical biomarkers related to cardiac tissue remodeling, oxidative stress, and inflammatory responses and associated clinical biomarkers are summarized in the heat map in Figure 7H. Change in the color intensity clearly indicates that the YDR co-treatment in the ISP-stimulated H9C2 cells varyingly modulated the expression of the analyzed clinical and non-clinical CH biomarkers (Figure 7H).

## 4. Discussion

Cardiovascular disease remains one of the most prominent causes of mortality throughout the world. Heart failure is defined as a deficiency of the heart’s ability to adequately pump blood in response to systemic demands. YDR is a traditional Indian Ayurvedic formulation prepared using mercury (Hg), gold (Au), and tin (Sn). Generally, YDR is used as a neurostimulator and acts as a catalyst with other Ayurvedic formulations to increase their therapeutic efficacy [35]. According to the compendium of classical Indian medicinal text, Ayurveda Sar Sangraha, YDR can also be prescribed for cardiac ailments [36].

Based on the physicochemical characterization, YDR was found to be heterogeneous in shape and size, containing large quantities of Hg, Sn, and O. Complementing these findings, ICP-MS analysis of the YDR particles showed the presence of Hg (9.8%) and Au (7.9%). Interestingly, the absence of Au peak in the EDX and in the ICP-MS analysis might be due to the masking effect induced by the gold-palladium coating process of the samples. The detected elemental constituents of the YDR were close to those mentioned in the ancient Indian text of *Bhaisajya Ratnāvalī* [35]. Hence, our ICP-MS detections were well within the prescribed range for Hg and Au. Dry powder XRD analysis showed the presence of HgS, in the forms of Cinnabar and Metacinnabar, pure Au, iron (hematite), As-Fe alloy, and arsenic impurity (As_4_O_6_). Mercury in the form of Cinnabar has been extensively used in traditional Indian and Chinese medicines in treating various diseases such as neuropathy and cardiac diseases [37,38]. Cinnabar has been shown to have low GI tract absorption as compared to mercuric chloride (HgCl_2_) and demonstrates 1000× lower toxicity as compared to methyl mercury [37]. Other compounds such as hematite and pure Au have also been reported to not be toxic to humans in low doses. Hydrodynamic diameter analysis of the YDR particles using the dynamic light scattering technique showed that in the presence of biomolecules present in cell culture media with 2% FBS, their relative size distribution reduced significantly. This change in size distribution of the YDR particles was due to the formation of a bio-corona surrounding, modifying their relative surface attributes.

In the present study, we applied erythromycin as an inducer of CH in the *D. rerio* as earlier studies have shown that this can induce CH [8]. Since *D. rerio* contains a two-chambered heart, directing the hypertrophy toward the left ventricle is not possible. In the present study, based on the ECG reading development of CH was observed through an increase in the amplitudes of the QRS wave. Furthermore, analysis of the Cornell product, Cornell voltage, Sokolow–Lyon, and Romhilt–Estes point criteria obtained from the ECG readout clearly showed the induction of CH [28]. Induction of CH was also visually confirmed through the histological analysis of the whole fish heart and septum wall thickness. Treatment of the ERY-stimulated fish with YDR or verapamil clearly showed an amelioration of the ERY-induced CH in the fish heart. Earlier studies using zebrafish have shown similar efficacy with the synthetic standard-of-care drug verapamil [39,40]. Elevated CRP and platelet aggregation levels have been associated with an increase in cardiovascular function failures in hypertensive adults [33,41]. Treatment of the CH-induced *D. rerio* with YDR significantly reduced the stimulated release of CRP and platelet aggregation, indicating a reduction in the level of CH. Our results confirmed the earlier ECG and oxidative stress-related findings and showed that the YDR has equal potential in ameliorating drug-induced CH. Therefore, this is the first study to report the efficacy of metal-based ancient Ayurvedic medicine in modulating CH.

Based on the in-vivo study results, we expected YDR to play a major role in cardioprotection, by reducing stress signaling, and cardiac remodeling. In our in-vitro studies, we applied isoproterenol (ISP) for the induction of hypertrophy. ISP helps in normalizing the diminished heartbeat. However, prolonged use of ISP could also lead to change in the left and right ventricular wall function and physiology inducing cardiomyopathy [42]. Therefore, in our present study, we employed ISP for inducing CH in the rat cardiomyocyte (H9C2) cell lines, through the process of oxidative stress-induced damage. Applying a sub-toxic inhibitory 20% (IC_20_) concentration of ISP in the H9C2 cells, we further studied the efficacy of YDR. ISP induced loss of cell viability, redox imbalance, and up-regulation in the mRNA expression of non-clinical and clinical biomarkers of CH. YDR given to the H9C2 cells in combination with the ISP ameliorated these drug-induced CH biomarkers.

Oxidative stress-induced redox imbalance in the cardiomyocytes is one of the primary reasons for the development of myocyte hypertrophy [43]. Mitochondria are responsible for the generation of ~95% of ATP required by the heart for functioning, cellular signaling, controlled cell death, and generation of ROS such as superoxide anions and hydrogen peroxide [44]. Hence, prolonged cardiac oxidative stress and development of CH is directly associated with mitochondrial dysfunction and the generation of oxidative stress. In our study, we observed that ISP induced an increase in mitochondrial membrane potential along with ROS and O^−2^ generation. Potential growth in the ROS generation leading to a reduction in the cardiac mitochondrial cytochrome c and ubiquinone levels further modulating the mitochondrial membrane potential has been well-documented [45,46]. H9C2 cells with a co-treatment of the ISP and YDR showed a significant reduction in oxidative stress through the reduced release of both intracellular ROS and O^−2^. Hence, YDR acted as an anti-oxidant, or as a stimulator for cellular anti-oxidant (catalase and superoxide dismutase) levels. Both these antioxidants were significantly reduced through the treatment of H9C2 cells with ISP alone and enhanced through a co-treatment with YDR.

Cyclooxygenases (COX) and nicotinamide adenine dinucleotide phosphate oxidases (NADPH oxidases/NOX) represent the primary group of enzymes involved in the generation of ROS. Overexpression of COX-2 has been associated with angiotensin-II induced oxidative stress signaling pathways and CH [47]. In the present study, co-exposure of the H9C2 cells with YDR and ISP led to an inhibition in the over-expression of COX-2 mRNA triggered by ISP alone. Both these biomarkers indicated that they inhibited the induction of CH in the H9C2 cells possibly through the inhibition of the AT1R pathway. As a part of the mitochondrial enzymatic system, NOX acts as the main source for the production of O^−2^ in cardiomyocytes [48]. NOX plays an important role in the pathogenesis of cardiac remodeling [49]. A steady overexpression of NOX-2 and NOX-4 were depicted in heart failure patients, and the deletion of NOX-4 was observed to inhibit 80% development of CH in rats [48]. In our mRNA expression study, YDR was found to significantly down-regulate the over-expression of NOX-2 and NOX-4 in the cardiomyocytes induced by ISP, single-handedly. The study suggested a significant role for the YDR to act as an antioxidant inhibiting the ISP-induced oxidation and hypertrophy.

Oxidative stress activates the cell-death-associated cascades and contributes to maladaptive myocardial remodeling. The induction of hypertrophy in the adult myocardium is associated with the partial recapitulation of the embryonic program observed during cardiac development. Atrial natriuretic factor (ANF) is associated with remodeling during the induction of CH. It is a very sensitive marker, and overexpression of ANF in adult heart can lead to the development of defects in the chamber myocardium of the atria, ventricles, atrioventricular canal, and pacemaker tissue patients [49]. In our study, we found that the YDR co-treatment with ISP reduced the mRNA over-expression of the ANF induced by ISP alone, indicating a reduction in hypertrophy conditions. Other molecular sources for measuring the CH inhibitory efficacy of YDR were clinical biomarkers, troponin I (TNN-I), troponin T (TNN-T), and cardiolipin (CRLS-1). Cardiac troponin I and T are subunits of the cardiac actin–myosin complex, and their release into the systemic circulation represents myocardial injuries and necrosis [50]. Cardiolipin (CRLS-1) is a phospholipid found only in the inner mitochondrial membrane and plays an important role in the formation of mitochondrial cristae and super-complexes in the electron transport chain and bioenergetics. Hence, any mitochondrial dysfunction in the cardiomyocytes is represented by cardiolipin release in circulation. In our study, we observed an increase in the mRNA expression level of TNN-I, TNN-T, and CRLS-1, by ISP induction. An increase in these levels confirmed the oxidative damage induced by the ISP and the onset of cardiomyocyte hypertrophy. YDR co-treatment led to a significant reduction in the mRNA expression levels of these three genes, indicating a reduction in cellular oxidative damage and inhibition of CH. Hence, the mRNA expression analysis validated the cellular and biochemical parameters observed for the antioxidant behavior of the YDR in modulating ISP-induced redox imbalance and cardiac hypertrophy. Results from the study indicate a noteworthy role of the YDR in modulating CH. Future studies would be directed towards understanding the intracellular role of metals associated with YDR in modulating cardiac hypertrophy; and exploring the possibility of clinical trials with YDR in patients with cardiac hypertrophy.

## 5. Conclusions

In conclusion, zebrafish represent an effective model for the study of cardiac hypertrophy and its amelioration by traditional Indian medicine Yogendra Ras (YDR). YDR was found to be formed mainly of cinnabar, gold, and tin. In the ERY-stimulated *D. rerio* CH model, YDR acted as a therapeutic medicine and helped in the normalization of cardiac parameters, represented by ECG, heart morphology, and septum wall thickness. YDR also ameliorated ERY-stimulated release of pro-inflammatory C-reactive protein and normalized platelet aggregation time. In the in-vitro model of ISP-stimulated H9C2 cells, YDR was found to act as an anti-oxidant agent, helping in the reduction of excessive intracellular reactive oxygen species and superoxide anions generation and an associated increase in mitochondrial membrane potential. YDR also reduced the ISP-stimulated changes in the levels of antioxidant enzymes and, at molecular levels, clinical and non-clinical biomarkers associated with CH. Taken together, the results suggest a role for YDR as an alternative therapeutic medicine for the treatment of oxidative stress and inflammation-associated cardiac ailments such as cardiac hypertrophy.

## Figures and Tables

**Figure 1 biomolecules-10-00600-f001:**
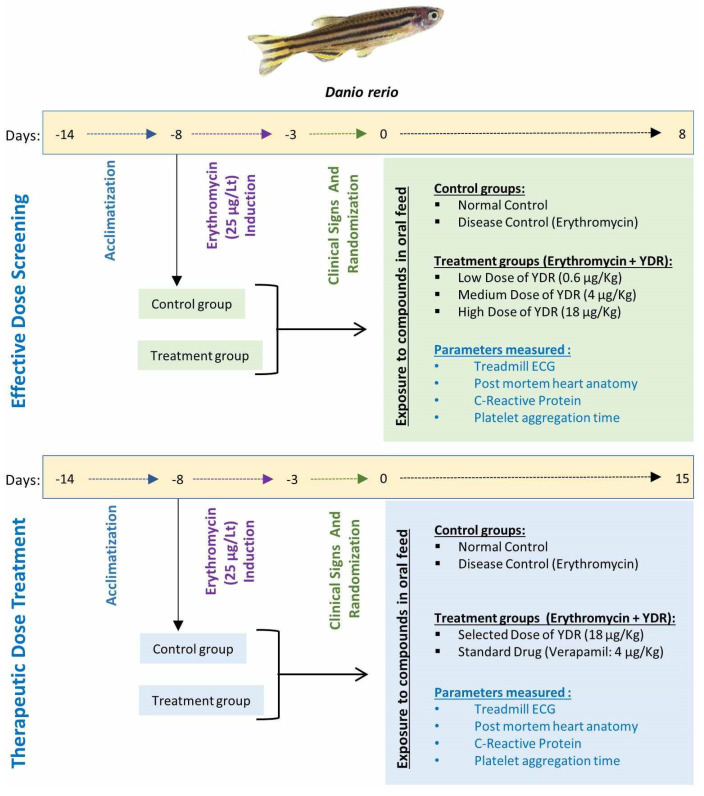
Experimental design of *D. rerio* exposure to cardiac hypertrophy induction by erythromycin. Duration of the effective dose study was 7 days, and for therapeutic dose study it was 14 days.

**Figure 2 biomolecules-10-00600-f002:**
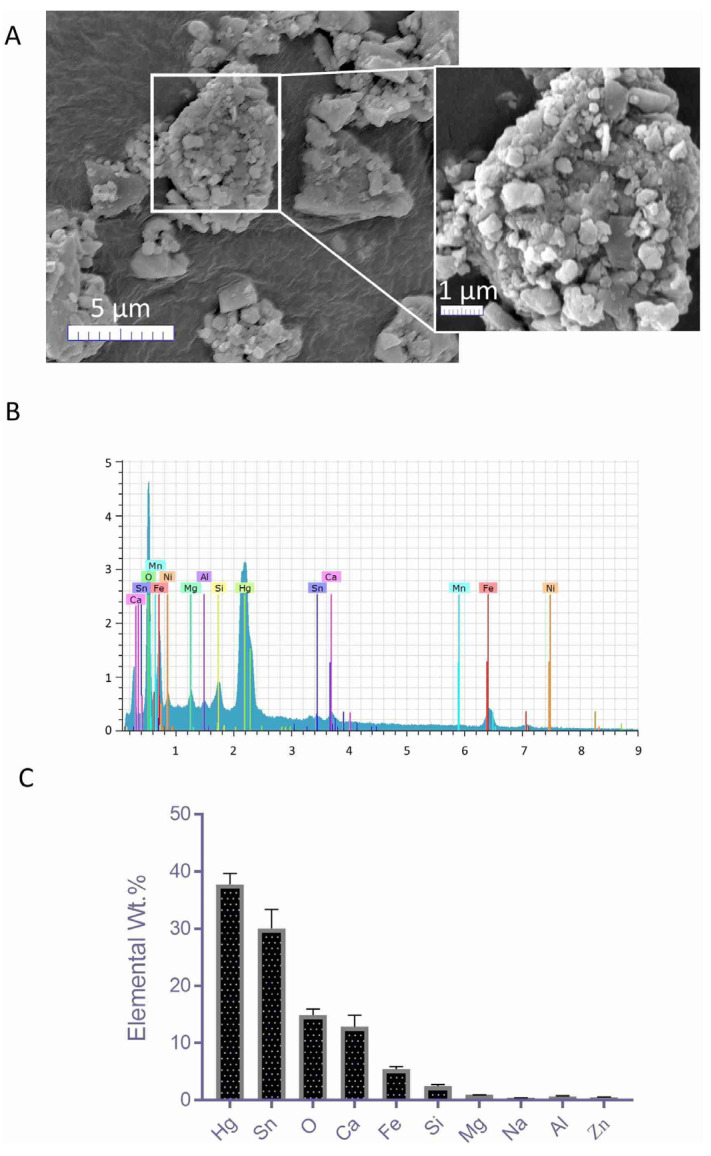
Scanning electron microscope (SEM) and electron dispersive X-ray (EDX) analysis of Yogendra Ras (YDR). (**A**) SEM analysis of the YDR powder showed the presence of a heterogeneous mass of particles of varying shapes and sizes. Particles were present individually or as aggregates. (**B**) Electron dispersive X-ray analysis showed the presence of various elements on the surface of the YDR particles. (**C**) Quantitatively highest elemental concentrations were detected for mercury (Hg), tin (Sn), oxygen (O), calcium (Ca) and iron (Fe). Several other elements such as silicon (Si), magnesium (Mg), sodium (Na), aluminum (Al), and zinc (Zn) were also detected but at ≤ 1% *w*/*w* concentrations.

**Figure 3 biomolecules-10-00600-f003:**
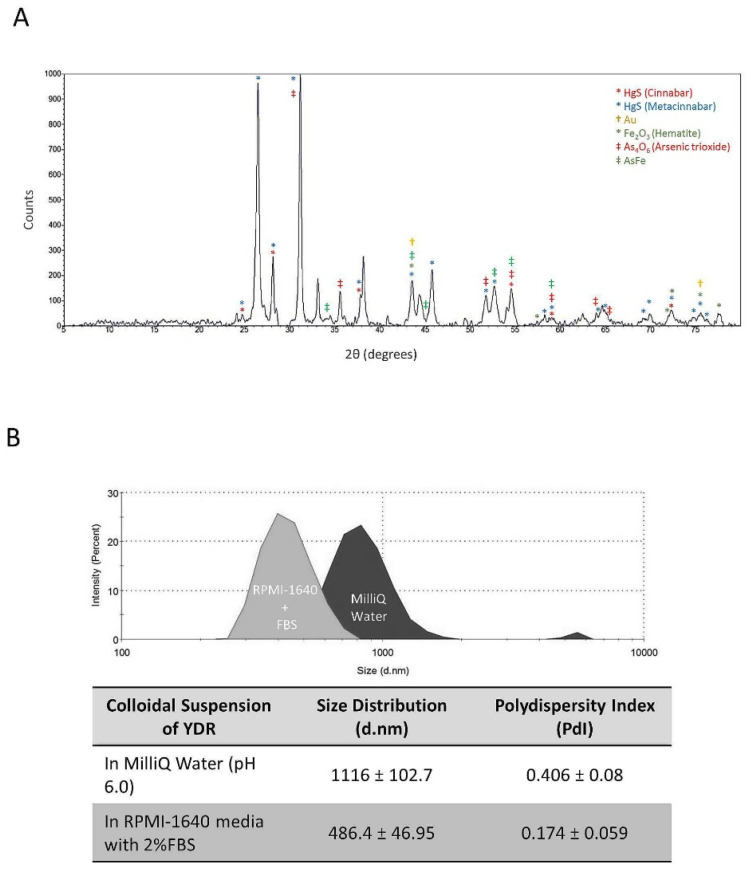
X-ray diffraction (XRD) and size distribution analysis of dry Yogendra Ras (YDR) sample. (**A**) XRD analysis showed the presence of cinnabar (HgS), metacinnabar (HgS), pure gold (Au), hematite (Fe_2_O_3_), arsenic trioxide (As_4_O_6_), and arsenic-iron alloy (AsFe). (**B**) Dynamic light scattering analysis of the YDR showed them to be present in a bimodal phase when suspended in pure MilliQ water. The size distribution showed the presence of particles with a hydrodynamic diameter of 1116 ± 102.7 d.nm along with a high polydispersity index (PdI) value representing an unstable colloidal suspension. When the YDR particles were suspended in cell culture media containing 2% FBS, the particle size distribution changed to a monomodal distribution showing the hydrodynamic diameter of 486 ± 46.95 d.nm and a lower PdI value representing a stable colloidal suspension.

**Figure 4 biomolecules-10-00600-f004:**
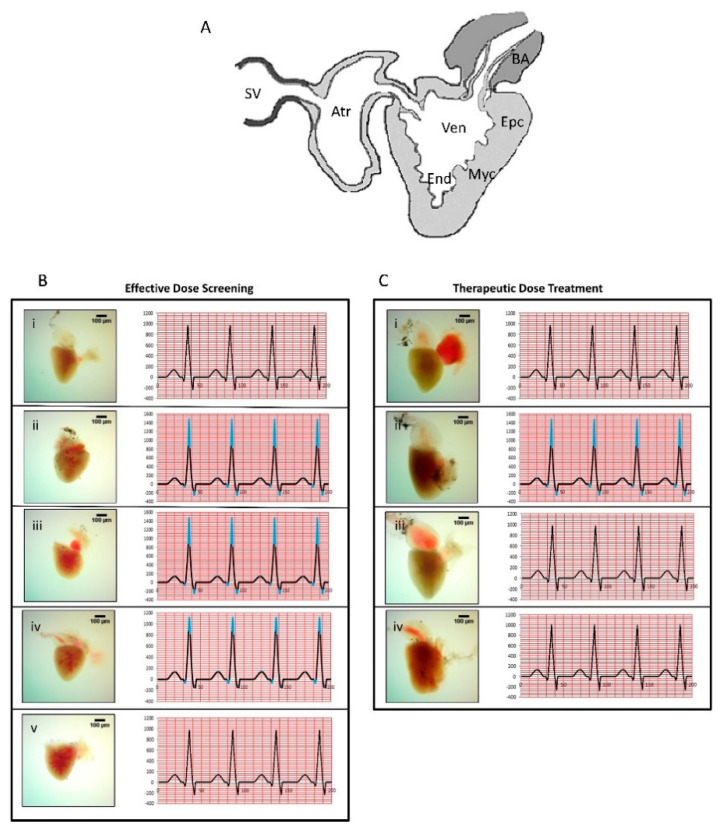
Electrocardiogram (ECG) analysis and whole heart histological analysis of erythromycin (ERY)-stimulated cardiac hypertrophy. (**A**) Schematic diagram of *Danio rerio* heart: sinus venosus = SV, atrium = Atr, ventricle = Ven, bulbus arteriosus = BA, epicardium = Epc, myocardium = Myc, endocardium = End (graphics adapted from Yacoub et al. [32]). (**B**) In the effective dose screening analysis, ECG and heart anatomy of the *D. rerio* were studied for (**i**) normal control; (**ii**) disease control (25 µg/Lt erythromycin (ERY)); (**iii**) low dose (0.6 µg/kg of YDR); (**iv**) medium dose (4 µg/kg of YDR); and (**v**) high dose (18 µg/kg of YDR). The results indicated induction of CH by ERY. No changes were detected in the CH parameters following the 7 days of effective dose treatment with YDR at the low and medium doses. Mild changes were observed at the highest tested dose of YDR. (**C**) In the therapeutic dose treatment study, *D. rerio* were grouped as (**i**) normal control; (**ii**) disease control (25 µg/Lt ERY); (**iii**) standard drug (4 µg/kg of verapamil); (**iv**) therapeutic dose of YDR (18 µg/kg of YDR). The results indicated that, following 14 days of treatment, both the verapamil and YDR successfully ameliorated the CH induced by ERY in *D. rerio*.

**Figure 5 biomolecules-10-00600-f005:**
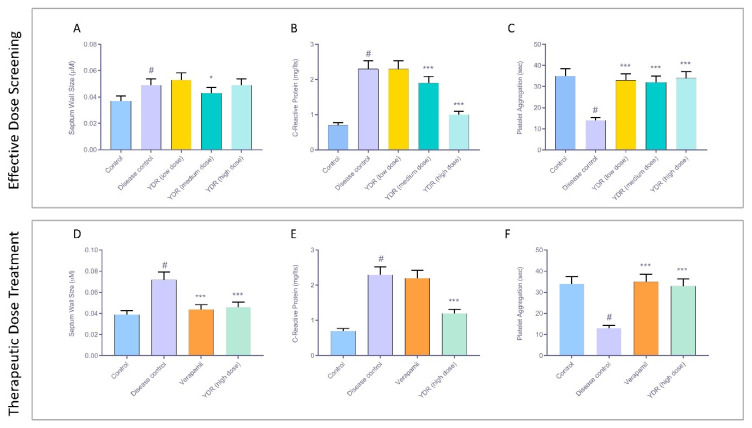
Biomarker analysis for effective dose screening and therapeutic dose treatment of Yogendra Ras (YDR) in *Danio rerio* stimulated with erythromycin. (**A**) Effective dose screening of YDR in *Danio rerio* showed no change in the septum wall thickness of the ERY-stimulated *D. rerio* compared to the disease control. (**B**) ERY stimulation of the *D. rerio* significantly increased the systemic release of the inflammatory C-reactive protein. This was reduced in a dose-dependent manner through the treatment of the ERY-stimulated fish. (**C**) ERY stimulation of the *D. rerio* significantly decreased the platelet aggregation time. YDR treatment of the ERY-stimulated *D. rerio* at varying concentrations recovered the platelet aggregation time back to normal. (**D**) Therapeutic treatment of the ERY-stimulated *D. rerio* with YDR (18 µg/kg) and standard of care drug, verapamil (4 µg/kg) significantly reduced the CH-associated increase in septum wall thickness. (**E**) Therapeutic treatment of the ERY-stimulated *D. rerio* with YDR (18 µg/kg) significantly reduced the release of the inflammatory C-reactive protein. Verapamil did not induce any quantitative change in the release of the inflammatory protein. (**F**) Therapeutic treatment of the ERY-stimulated *D. rerio* with YDR (18 µg/kg) significantly recovered the stimulated platelet aggregation time back to the normal level. *n* = 24. One-way ANOVA followed by Dunnett’s post-hoc test was applied to study statistical significance: normal control versus disease control (# *p*-value <0.001) and YDR/verapamil treatment versus disease control (* *p*-value < 0.05; *** *p*-value < 0.001).

**Figure 6 biomolecules-10-00600-f006:**
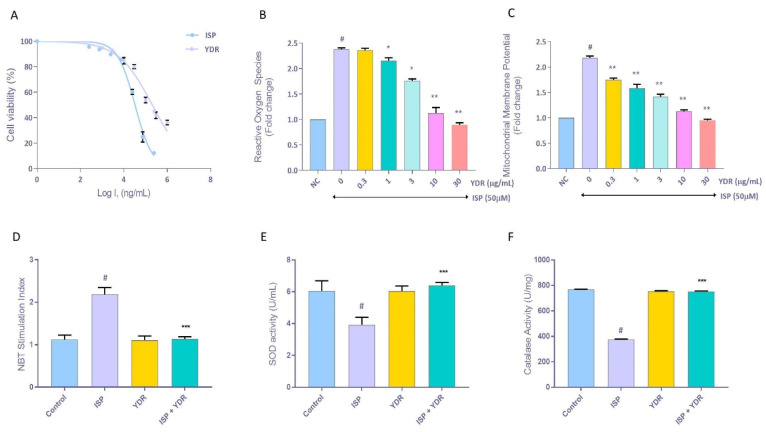
Dose screening and oxidative stress analysis in H9C2 cells exposed to isoproterenol (ISP) and Yogendra Ras (YDR). (**A**) MTT-based dose screening in the H9C2 cells showed a dose-dependent loss of cell viability following exposure to the ISP (IC_20_ = 50 µM and IC_50_ = 122.4 µM) and YDR (IC_20_= 30 µg/mL and IC_50_ = 205 µg/mL). (**B**) Stimulation of the H9C2 cells with ISP (50 µM) induced up-regulation in the production of intracellular reactive oxygen species and its reduction following treatment with YDR. (**C**) Mitochondrial membrane potential (MMP) increment was detected in the ISP-stimulated H9C2 cells. It was reduced following co-treatment with YDR. (**D**) YDR also reduced the ISP-stimulated generation of superoxide ions (O^−2^) in H9C2 cardiomyocytes as indicated through the intracellular formation of formazan in the NBT assay. (**E**) Reduction in the generation of O^−2^ in the ISP-stimulated H9C2 cells following treatment with YDR was confirmed through the recovery in the intracellular levels of superoxide dismutase (SOD) enzyme. (**F**) Intracellular catalase enzyme levels were also recovered in the ISP-stimulated H9C2 cells following treatment with YDR. All the experiments were performed in biological and technical triplicates. Results represent mean ± SD. The statistical analysis results were analyzed using one-way ANOVA followed by Dunnett’s post-hoc test: *p*-value # < 0.01 (ISP alone versus normal control) and ** < 0.01, *** < 0.001 (YDR co-treatment versus ISP alone).

**Figure 7 biomolecules-10-00600-f007:**
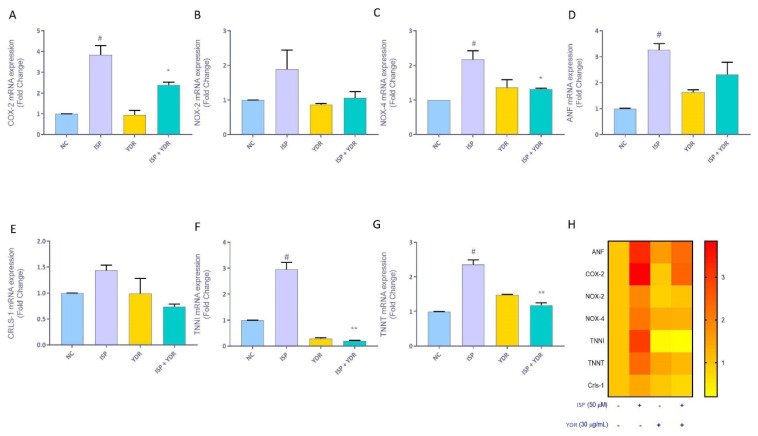
Down-regulation of cardiac-hypertrophy-associated non-clinical and clinical molecular biomarkers by Yogendra Ras (YDR) in the isoproterenol (ISP)-stimulated H9C2 cells. H9C2 cells stimulated with ISP showed an increase in the mRNA expression levels of the pro-inflammatory mediator genes: (**A**) Cyclooxygenase-2 (COX-2), (**B**) NADPH oxidase-2 (NOX-2). and (**C**) NADPH oxidase-4 (NOX-4) using quantitative real-time PCR. Significant down-regulation of these pro-inflammatory genes was observed following a co-treatment of the ISP-stimulated H9C2 cells with YDR. Treatment of the H9C2 cells with ISP also induced an up-regulation of the expression of the fetal cardiac fetal gene (**D**) atrial natriuretic factor (ANF) and (**E**) cardiolipin (CRLS-1), (**F**) troponin I (TNN-I), and (**G**) troponin T (TNN-T). Expression of all the genes was normalized against the house-keeping gene β-actin. Co-treatment of the ISP-stimulated H9C2 cells with YDR significantly reduced the expression of all the up-regulated non-clinical and clinical genes. H) The heat map was generated to summarize the gene expression study. All the experiments were performed in biological and technical triplicates. Results represent mean ± SD. For the statistical analysis, results were analyzed using one-way ANOVA followed by Dunnett’s post-hoc test: *p*-value # < 0.01 (ISP/YDR versus normal control) and * < 0.05, **< 0.01 (YDR co-treatment versus ISP alone).

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
