# Peer review of "Application of Zebrafish Model in the Suppression of Drug-Induced Cardiac Hypertrophy by Traditional Indian Medicine Yogendra Ras"

_biomolecules, 2020, doi:10.3390/biom10040600_

Round 1

Reviewer 1 Report

I thoroughly enjoyed reviewing this publication and it appears to be very well designed and rich in evidence that YDR should be considered for main stream treatment of cardiac disease/failure. 

One important area for improvement would be referencing the requirements of the ARRIVE guidelines. https://www.nc3rs.org.uk/arrive-guidelines

I would not be able to recreate some of the experiments using zebrafish due to lack of husbandry parameters. I could not easily see the numbers of fish that you used. 

Specific comments relating to this

Line 36 - Sentence starting further is not clear. Please re-write.

Line 39 - change to represent humans closely

Line 110 - breeding history, strain. refer to ARRIVE for ideas. 

Other comments

Line 19 - remove with

Line 25 - not a capital Z

Lines 62-74 - did your YDR have Aloe vera as a binding agent? This would have been a good additional control.

Lines 120-129 - saline controls would have been good as well. 

Line 136 Which company made the pellets?

Line 139 Ask yourself if I could repeat the feeding regime? Estimated dosing seems vague. Dosing? You cannot be sure of the amount that they ate? Did you watch them finish the food each day? 

Line 166 no need for set. Sedation time? Were these fish recovered or euthanised immediately?

Line 187 cell culture media supplier is missing

Line 280 I am wondering what state the O is in and if it is available to the cell?  To discuss?

Lines 282-289 I would prefer the chemical names in the text rather than in the figure legend but this maybe a word count issue?

Line 337-349 Figure 4. I cannot understand the majority of your reported results for the heart structure and appearance. I require labelled diagrams to help explain this. Also the n number is not obvious to me? 

Line 371 The presence of verapamil is not detailed in the figure legend

Line 508 Figure 7, H Should say YDR  not HAV?

Line 532 edit "in"

Wondering if the discussion could reflect upon the presence of these elements cellularly? Potential for clinical trials etc?

Author Response

I thoroughly enjoyed reviewing this publication and it appears to be very well designed and rich in evidence that YDR should be considered for main stream treatment of cardiac disease/failure. 

Response: We sincerely thank the esteemed reviewer for the kind and encouraging words regarding the scientific quality of our manuscript. We have closely followed the suggestions provided by the reviewer and have updated the manuscript accordingly.

One important area for improvement would be referencing the requirements of the ARRIVE guidelines. https://www.nc3rs.org.uk/arrive-guidelines

Response: We appreciate the suggestion provided by the reviewer and have checked compliance of our study to ARRIVE guidelines. We have also added to the relevant ARRIVE reference.

I would not be able to recreate some of the experiments using zebrafish due to lack of husbandry parameters. I could not easily see the numbers of fish that you used. 

Response: We show our gratitude to the esteemed reviewer for the comment. Under sub-heading ‘2.3 Experimental Animals: D. rerio maintenance’ we have mentioned the animal husbandry parameters, that will help in the replication of the experiments performed in the current manuscript. We also thank the reviewer for pointing towards the discrepancy in the number of fish used in the current study. We have now corrected the number of fish used in the current study as, Line 118 ‘A total of 24 216 adult male D. rerio with a bodyweight of 0.5 g and length of 25-30 mm were selected for this study’, and in Line 123 ‘Fish were divided into 9 groups containing 24 fish per group’.

Specific comments relating to this

Line 36 - Sentence starting further is not clear. Please re-write.

Response: We highly appreciate the suggestion and deleted the sentence, ‘Further the small size of D. rerio helps in better management and handling of storage space required for the fish’ from the manuscript as we feel it does not fit the present manuscript context.

Line 39 - change to represent humans closely

Response: We acknowledge the correction suggested by the reviewer and have made the correction in the manuscript.

Line 110 - breeding history, strain. refer to ARRIVE for ideas.

Response: Thank you for the query. Since the fish were obtained from an external CRO, we acknowledge that it is difficult to obtain the breeding history.

Other comments

Line 19 - remove with

Response:  Thank you very much. ‘With’ has been removed.

Line 25 - not a capital Z

Response: Thank you, it has been corrected.

Lines 62-74 - did your YDR have Aloe vera as a binding agent? This would have been a good additional control.

Response: We agree that Aloe vera could have been an interesting control. As per our calculations, the amount of Aloe vera present as a binder in Yogendra ras is ~2% w/w. This would be ~5 mg/day human dosage. As per Sumi et al., 2019, 5% feed supplement with Aloe vera helped in resisting isoprenaline induced cardiac hypertrophy in rats. In our future planned studies on Yogendra ras, involving larger animals we will definitely explore Aloe vera extract used as binder as an additional control.

Reference: Sumi FA, Sikder B, Rahman MM, Lubna SR, Ulla A, Hossain MH, Jahan IA, Alam MA, Subhan N. Phenolic Content Analysis of Aloe vera Gel and Evaluation of the Effect of Aloe Gel Supplementation on Oxidative Stress and Fibrosis in Isoprenaline-Administered Cardiac Damage in Rats. Prev Nutr Food Sci. 2019; 24(3):254-264. doi: 10.3746/pnf.2019.24.3.254.

Lines 120-129 - saline controls would have been good as well. 

Response: Thank you very much for identifying the missing information. We have now added in the manuscript, ‘In the normal control fish, equivalent volume of 0.9% saline solution was added to the housing water. Normal control and Erythromycin exposed fish were observed and stabilized from day 5 onwards for the initiation of YDR/ Verapamil treatments’.

Line 136 Which company made the pellets?

Response: The fish were fed with standard tetraMin® flakes obtained from Tetra, VA. Yogendra ras was mixed with the TertaMin® flakes at known concentration and was extruded into uniform pellets.

Line 139 Ask yourself if I could repeat the feeding regime? Estimated dosing seems vague. Dosing? You cannot be sure of the amount that they ate? Did you watch them finish the food each day? 

Response: All the zebrafish were fed on a 24 hrs cycle, i.e. the fish were starved for 24 hrs. Individual fish were isolated and fed with an estimated number of pallets. Their feeding was observed by a technician and the fish were kept in their individual feeding tank until they completely fed and left nothing of the pallet.

Line 166 no need for set. Sedation time? Were these fish recovered or euthanised immediately?

Response: We recognize the interesting point reviewer has raised. Sedation of the fish at 14°C was necessary to minimize their suffering during the platelet aggregation study. It was not possible to use a euthanized fish at 4°C for studying the platelet aggregation parameter as it would lead to superficial platelet damages. Euthanization at 4°C was performed immediately, Therefore, we have modified the sentence as, ‘Post measurement of the platelet aggregation, all the study fish were euthanized immediately using ice-cold water set at 4°C’.

Line 187 cell culture media supplier is missing

Response: We have deleted the incomplete sentence, ‘Cell culture media’. Supplier for the cell culture media Dulbecco’s Modified Eagle Medium (DMEM) was Thermofisher Scientific Inc., USA mentioned in line 89.

Line 280 I am wondering what state the O is in and if it is available to the cell?  To discuss?

Response: We appreciate the query put forward by the reviewer. Unfortunately, using the electron dispersive X-ray (EDX) analysis we could only analyze the elemental composition of YDR and not the O speciation. However, based on the X-ray diffraction analysis, the source of oxygen was detected as hematite (Fe2O3) and arsenic trioxide (As2O3). In the cell culture experiments, YDR was given the cells in the form of an unstable colloidal suspension prepared in cell culture media as determined using the dynamic light scattering technique. Hence, YDR formulation was available to the cells with the passage of time following its sedimentation to the bottom of the cell culture flask. We agree that the speciation of O is important in the induction of oxidative stress in the cells, but it is difficult to speculate on it in case of a complex formulation like YDR.    

Lines 282-289 I would prefer the chemical names in the text rather than in the figure legend but this maybe a word count issue?

Response: We agree with the suggestion of the esteemed reviewer and have added the chemical names along with their chemical symbols.

Line 337-349 Figure 4. I cannot understand the majority of your reported results for the heart structure and appearance. I require labelled diagrams to help explain this. Also the n number is not obvious to me? 

Response: We thank the esteemed reviewer. have added an additional labelled zebrafish schematic figure as figure 4a.

Line 371 The presence of verapamil is not detailed in the figure legend

Response: We thank the reviewer. Details regarding the Verapamil is mentioned under figure 4B as ‘iii) Standard drug (4 µg/kg of Verapamil + ERY)’. We have also mentioned in the therapeutic dose, ‘). The results indicated that following 15 days treatment, both the Verapamil and YDR successfully ameliorated the CH effected induced by ERY in D. rerio’.

Line 508 Figure 7, H Should say YDR not HAV?

Response: We thank the reviewer for pointing out the error. We have corrected it now.

Line 532 edit "in"

 Response: Thank you. It has been edited.

Wondering if the discussion could reflect upon the presence of these elements cellularly? Potential for clinical trials etc?

Response: We thank the esteemed reviewer for the wonderful suggestion, and added in the discussion part Line 670 ‘Results from the study indicate a noteworthy role of the YDR in modulating CH. Future studies would be directed towards understanding the intracellular role of metals associated with YDR in modulating the cardiac hypertrophy and exploring the possibility of clinical trials with YDR in patients with cardiac hypertrophy’.

Reviewer 2 Report

The manuscript by Balkrishna et al. studies a zebrafish model of cardiac hypertrophy, showing that the traditional Indian medicine Yogendra Ras mitigates the cardiac hypertrophy processes.  They also validate their findings using a murine cardiomyocyte line.  Overall, I was very impressed with the manuscript and its completeness.  I have some minor concerns that are outlined below.  One major concern is that the text in the figures is too small to see clearly (try to read the text in a printed page).

1- In the abstract, line 11, “Zebrafish… employed in the pathological…” should be changed to “Zebrafish… employed to model the pathological…”.

2- The methods has redundancy in description of the murine cell line.  It is not necessary to repeat the plating density and the treatment conditions several times.

3- Cornell is spelled wrong (Corenell) repeatedly.

4- Why is the high dose effective in Fig 5D but not in Fig 5A?

5- On line 426, Fig 4F should read Fig 5F.

6- The discussion repeats parts of the introduction and results.  Reiteration of the results in not necessary.  The discussion should be more focused, and it should be shortened by reducing the repetition and reiteration.

Author Response

The manuscript by Balkrishna et al. studies a zebrafish model of cardiac hypertrophy, showing that the traditional Indian medicine Yogendra Ras mitigates the cardiac hypertrophy processes.  They also validate their findings using a murine cardiomyocyte line.  Overall, I was very impressed with the manuscript and its completeness.  I have some minor concerns that are outlined below.  One major concern is that the text in the figures is too small to see clearly (try to read the text in a printed page).

Response: We sincerely thank the esteemed reviewer for the kind and encouraging words regarding the scientific quality of our manuscript. We have provided response to the suggestions made by the reviewer below and have updated the manuscript accordingly.

1- In the abstract, line 11, “Zebrafish… employed in the pathological…” should be changed to “Zebrafish… employed to model the pathological…”.

 Response: Thank you very much We have modified the line 11 as per reviewer suggesiton.

2- The methods has redundancy in description of the murine cell line.  It is not necessary to repeat the plating density and the treatment conditions several times.

 Response: We appreciate the reviewer for the keen observation. We have now removed redundancy from the materials and methods.

3- Cornell is spelled wrong (Corenell) repeatedly.

 Response: We have now performed the spelling correction throughout our manuscript.

4- Why is the high dose effective in Fig 5D but not in Fig 5A?

 Response: We show our gratitude to the reviewer for the query. In both the figure 5A and 5D the negative control zebrafish septum wall thickness was found to be 0.04 µm and in the erythromycin pre-treated zebrafish following treatment with the highest dose of YDR the septum wall thickness was determined at 0.05 µm. Hence, there is no variation in the relative values of septum wall thickness in the effective and therapeutic dose treatment of the zebrafish models treated with erythromycin. 

5- On line 426, Fig 4F should read Fig 5F.

Response: We thank the reviewer for pointing the error It has been corrected. 

6- The discussion repeats parts of the introduction and results.  Reiteration of the results in not necessary.  The discussion should be more focused, and it should be shortened by reducing the repetition and reiteration.

Response: We thank the esteemed reviewer for the comment and have removed repetition and reiterations from the discussion part of the manuscript. However, removing all the results from the discussion part was not possible as they are necessary to put forward the evidence for the efficacy of Yogendra ras as a cardiac hypertrophy ameliorating medicine and propose the mode of action behind its activity.
